# Relationship between saturation and contrast during internal limiting membrane peeling in 3D digital vitreoretinal surgery: A retrospective study

Saaya Fukase[1], Yoshihito Sakanishi[1]*, Keiji Matsumoto[1], Ayumi Usui-Ouchi[1],
Shintaro Nakao[2], Nobuyuki Ebihara[1]

1 Department of Ophthalmology, Juntendo University Urayasu Hospital, Urayasu, Chiba, Japan,
2 Department of Ophthalmology, Juntendo University, Tokyo, Japan

* ysakani@juntendo.ac.jp

## Abstract

This retrospective comparative study aimed to investigate the relationship between saturation and contrast when using Brilliant Blue G (BBG) staining in internal limiting membrane (ILM) peeling during vitreoretinal surgery using the NGENUITY® 3D Visual System (NGENUITY), a three-dimensional digital monitor. Participants were individuals undergoing vitreoretinal surgeries involving ILM peeling with BBG staining, performed by the same surgeon at Juntendo University Urayasu Hospital, from October 2022 to June 2023, using the NGENUITY system. After ILM peeling, the contrast ratio between the ILM-peeled area and stained area was calculated using normal color tone (Normal group) and color tone set to yellow, with saturation set to 0 (Monochrome-0 group) or 20 (Monochrome-20 group). The visibility of retinal hemorrhage was also examined. The findings were compared among the groups. The contrast ratios of surgical screens with different color settings were used as the main outcome measure. We included 27 patients (27 eyes; 16 female individuals; age: $68.9 \pm 9.8$ years). The contrast ratios were $1.62 \pm 0.15$, $2.05 \pm 0.44$, and $2.00 \pm 0.51$ in the Normal, Monochrome-0, and Monochrome-20 groups, respectively. The contrast ratios were significantly higher in the Monochrome-0 and Monochrome-20 groups than in the Normal group (both $p < 0.01$), whereas these ratios were similar in the Monochrome-0 and Monochrome-20 groups ($p = 0.554$). Although distinguishing retinal hemorrhage was difficult in the Monochrome-0 group, this distinction was possible in the Monochrome-20 group. In BBG staining of the ILM, using the monochromatic mode with the NGENUITY system improves contrast visibility; however, using slight saturation facilitates distinction of retinal hemorrhage and allows safe ILM peeling.

**Data availability statement:** All relevant data are within the paper and its Supporting Information files.

**Funding:** The author(s) received no specific funding for this work.

**Competing interests:** The authors have declared that no competing interests exist.

## Introduction

In intraocular surgery, a microscope is used with a three-dimensional (3D) camera attachment, rather than a standard operating microscope, and surgery can be performed while viewing the high-resolution 3D monitor. This type of heads-up or digitally assisted surgery has become popular in recent years, as it has various advantages [1]. First, the surgeon's posture is relaxed and the burden on the body and neck is reduced [2] as the surgical field is observed on the monitor. Second, the staff in the operating room can view the same image as the surgeon, facilitating information sharing and teaching [3]. Third, as the image is digital, the image resolution can be refined, phototoxicity can be minimized by reducing light exposure [4,5], surgical guidance can be displayed on the monitor during surgery, and color filters can be used to adjust the color tone and contrast, improving visibility [6,7]. For example, in cataract surgery using the NGENUITY® 3D Visual System (NGENUITY; Alcon AG, Geneva, Switzerland), a 3D digital monitor, the contrast ratio can be increased, improving visibility, by setting the image saturation to 0, making the image monochromatic, when creating the continuous curvilinear capsulorrhexis [8].

One of the macular operations in vitreoretinal surgery is internal limiting membrane (ILM) peeling, which is a particularly delicate procedure. This operation involves staining the ILM to make it easier to see, and then peeling it off; however, the visibility at this time influences the ease of performing the procedure. Imai et al. have reported that, using monochromatic images, as described above, makes it easier to create contrast during ILM peeling [9]. However, in their report, the ILM was stained using indocyanine green (ICG), which has been reported to be toxic in recent years [10], with many authors thereafter reporting using Brilliant Blue G (BBG) to stain the ILM [11,12]. However, if the ILM is stained with BBG and surgery is performed using monochromatic images with 0 saturation, retinal hemorrhage is difficult to detect, posing a safety concern. We hypothesized that the addition of a small amount of saturation, which changes the vividness of color, might help detect hemorrhage, making macular operations safer.

Thus, in this study, we investigated the relationship between saturation and contrast when using BBG in ILM peeling during vitreous surgery using the NGENUITY system.

## Materials and methods

This retrospective study was conducted in accordance with the Declaration of Helsinki. In addition, the ethics committee of the Juntendo Clinical Research and Trials Center approved the study (approval number, E21-0180). We accessed the medical records on November 2, 2023, for research purposes, and all extracted patient data were anonymized for analysis. Patient consent was obtained through an opt-out process approved by the ethics committee. Patients were informed about the research through the hospital homepage and could withdraw consent at any time. Written informed consent was waived due to the retrospective observational nature of the study, as approved by the ethics committee.

The participants of this study were non-consecutive patients (aged 20–80 years) without any history of vitreous surgery, undergoing vitreoretinal surgeries at Juntendo

University Urayasu Hospital between October 2022 and June 2023, performed by the same surgeon using the NGENUITY system, and in which the ILM was stained using BBG. The surgical instrument used was Alcon's Constellation, and the surgery was performed using a 4-port system with a vitreous hand light and a Synergetics chandelier light for illumination. NGENUITY version 1.4 was used, and the camera aperture was set to 50%. Additionally, the main image quality setting was a gain of vitreous 1. This setting was standardized for all surgeries. During surgery, the vitreous was first removed using a vitreous cutter, and 0.25 mg/mL of BBG was then sprayed onto the retinal surface to perform ILM peeling. The contrast ratio between the ILM-peeled area and the ILM-stained area was calculated.

The NGENUITY images were set to three different color settings: normal image mode (Normal group), monochrome image mode (Monochrome-0 group), and monochrome plus saturation image mode (Monochrome-20 group). In the Monochrome-0 and Monochrome-20 groups, the color channel was set to yellow, and the saturation was set to 0 and 20, respectively. Yellow–Blue is the channel value; the lower the blue value, the higher the yellow value. Images of the same patient with different settings are shown in Fig 1.

The contrast ratio was calculated by two evaluators using the Mac application "Contrast," and the average of their calculations was used for analysis. The contrast ratios were calculated for each group and were compared among the three groups by using the Friedman test with post-hoc Bonferroni correction. Data were analyzed using the Statistical Package for the Social Sciences (version 26; SPSS Inc., Chicago, IL, USA). The significance level was set to $p < 0.05$.

## Results

Twenty-seven patients (16 female individuals; age $68.9 \pm 9.8$ years; 27 eyes: 18 eyes with epiretinal membrane, 5 eyes with macular hole, 4 eyes with macular traction syndrome) were included. The contrast ratios were $1.62 \pm 0.15$ in the Normal group, $2.05 \pm 0.44$ in the Monochrome-0 group, and $2.00 \pm 0.51$ in the Monochrome-20 group. The contrast ratios were significantly higher in the Monochrome-0 and Monochrome-20 groups than in the Normal group ($p < 0.01$ for both) but were not significantly different between the Monochrome-0 and Monochrome-20 groups ($p = 0.554$) (Fig 2).

In cases with retinal hemorrhage, the contrast ratio between hemorrhagic and non-hemorrhagic areas in the Normal group was 1.51, whereas in the Monochrome-0 group, the contrast ratio was 1.16, making it difficult to distinguish the boundaries between hemorrhagic and non-hemorrhagic areas. However, in the Monochrome-20 group, in which slight

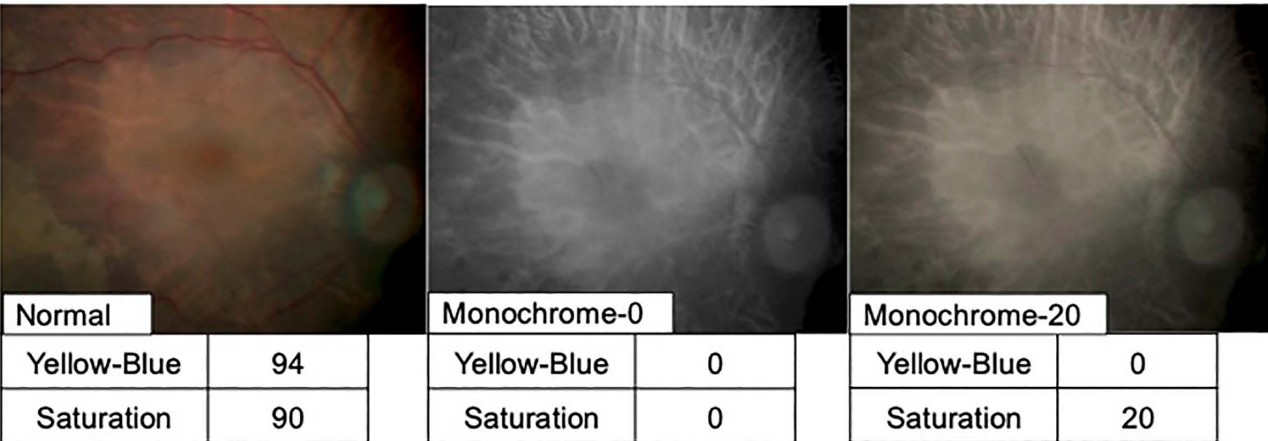

**Fig 1. Images of the same case under each image quality setting.** Yellow–Blue is the channel value, and as the blue value decreases, the yellow value increases. Saturation represents the vividness of the color. The following channels were compared in each captured image: Normal [Yellow–Blue 94, Saturation 90]; Monochrome-0 [Yellow–Blue 0, Saturation 0]; and Monochrome-20 [Yellow–Blue 0, Saturation 20].

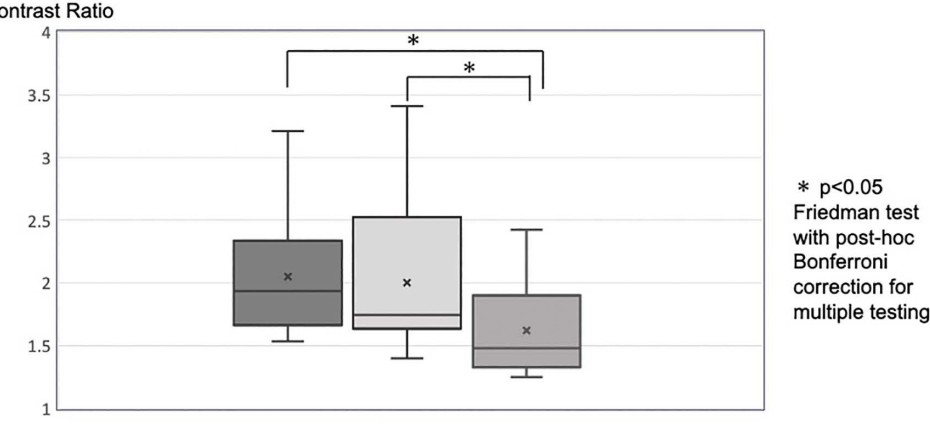

**Fig 2. The relationship between each color tone and contrast ratio.** A comparison of the contrast ratio between the Brilliant Blue G (BBG)-stained internal limiting membrane (ILM) and ILM-peeled area was conducted among the three-color tones. The contrast ratios were 1.62±0.15 in the Normal group, 2.05±0.44 in the Monochrome-0 group, and 2.00±0.51 in the Monochrome-20 group. The contrast ratios were significantly higher in the Monochrome-0 and Monochrome-20 groups than in the Normal group (p<0.01 for both), while no significant difference was observed between the Monochrome-0 and Monochrome-20 groups (p=0.554).

saturation was included, the contrast ratio between hemorrhagic and non-hemorrhagic areas increased to 1.31, facilitating their distinction as compared to that in the completely monochromatic mode (Fig 3).

## Discussion

In this study, we measured the contrast ratio in BBG staining of the ILM and found that the Monochrome groups had a significantly higher contrast ratio, compared with the Normal group. Using a completely monochromatic setting, distinguishing retinal hemorrhage was difficult; however, in a monochromatic setting with slight saturation, distinguishing hemorrhage became easier.

Previous studies have reported that using the monochromatic mode in 3D vitreoretinal surgery makes it easier to achieve contrast during ICG staining and peeling [9]; however, the use of ICG has been suggested as a possible cause of retinal damage [10]. Meanwhile, BBG has been reported to have a neuroprotective effect [13,14]. Additionally, when BBG

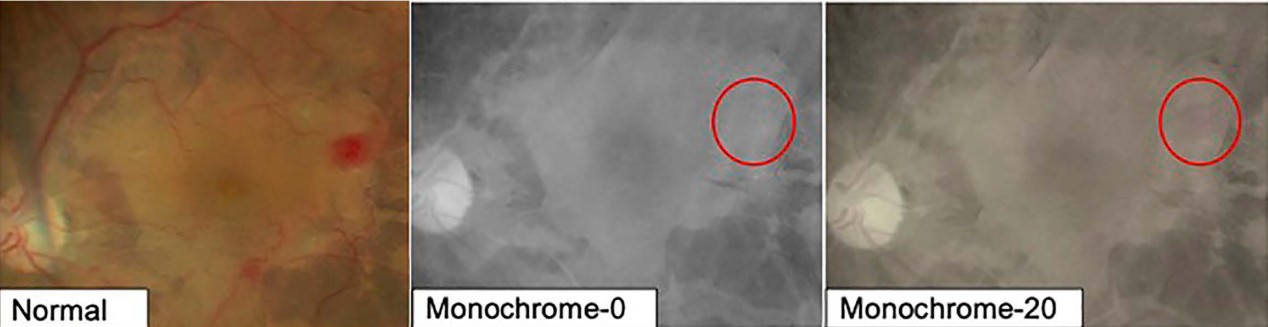

**Fig 3. Image of retinal hemorrhage.** By increasing the saturation, the contrast ratio between hemorrhagic and non-hemorrhagic areas was higher in the Monochrome-20 group than in the Monochrome-0 group. Thus, the use of color facilitated the distinction of the hemorrhagic area, compared with the use of the completely monochromatic mode.

was used in ILM peeling for macular holes, visual prognosis was reported to be better than that when ILM peeling was performed with ICG staining [15,16]. Therefore, we believe that this study of ILM staining using BBG holds significance in terms of clinical practice.

Color is a combination of three elements: hue (color tone), saturation (vividness), and brightness. Among these, adjusting the saturation affects the vividness of a color, whereas eliminating it completely will result in a black and white image, which is expressed only in terms of brightness, resulting in a so-called monochromatic state. Concurrently, all colors can be expressed by mixing red (R), green (G), and blue (B), which are considered to be the three primary colors of light. In terms of color tone, blue and yellow are complementary colors, and by setting the screen to yellow, blue will become darker. Among the three attributes of color, i.e., hue, saturation, and brightness, setting saturation to 0 to create a monochromatic image will lead to distinction of colors only by their brightness. By setting saturation to 0, all colors will be expressed as gray; however, the hue, saturation, brightness or HSB conversion formula can convert color to brightness (brightness of gray), which is expressed as ([maximum value of R, G, and B + minimum value of R, G, and B] ÷ 255) ÷ 2. In a normal-mode image of a patient with retinal hemorrhage, the (R, G, B) values of the hemorrhagic area were (113, 39, 27). When this image was converted to the monochromatic mode with saturation 0, the HSB conversion formula was (113 + 27) ÷ 255 ÷ 2 = 0.274, which implies that the brightness was 27.4% (Fig 4). The (R, G, B) values of the non-hemorrhagic area were (121, 71, 31), and when converted to the monochromatic mode with the HSB conversion formula, the brightness was (121 + 31) ÷ 255 ÷ 2 = 0.298, which means that the brightness was 29.8% (Fig 4). The brightness of the hemorrhagic and the non-hemorrhagic area, or the gray color density, was 27.4% and 29.8%, respectively, which represented a small difference. This explains why, even if the presence or absence of bleeding can be confirmed in the normal

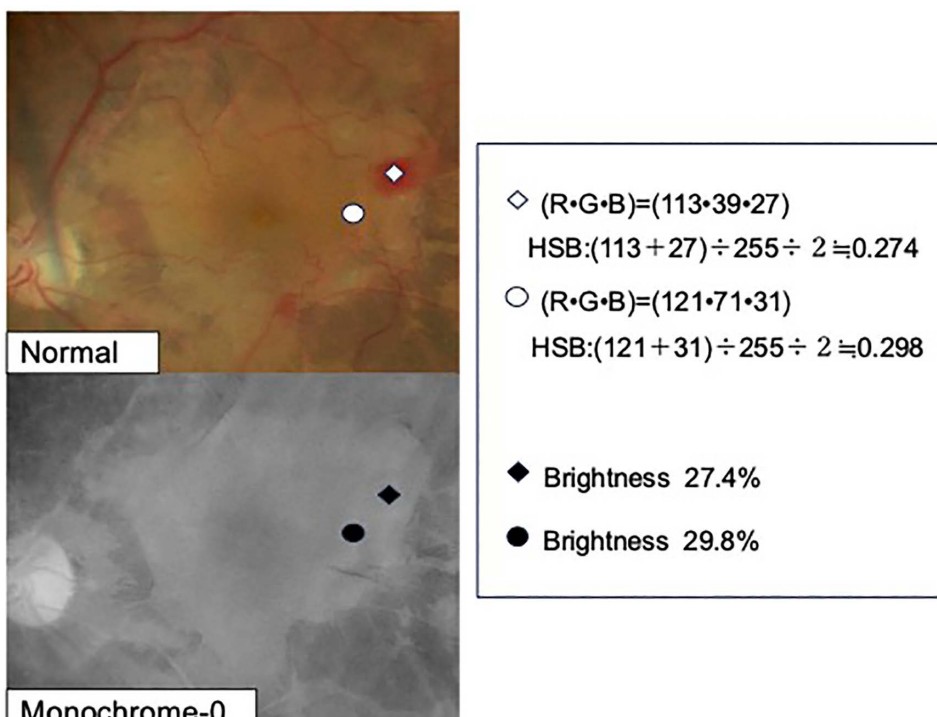

**Fig 4. Color component values of the hemorrhagic and non-hemorrhagic areas were substituted into a conversion formula for brightness, called the hue, saturation, brightness (HSB) conversion formula.** When the hemorrhagic area was converted to brightness only in the Normal mode, the brightness was 27.4%, whereas that of the non-hemorrhagic area was 29.8%. In other words, the difference in brightness between hemorrhagic and non-hemorrhagic areas in the Monochrome 0 mode, where saturation is 0, is small.

mode, when the saturation is set to 0 and the mode is set to completely monochromatic, the retinal hemorrhagic and non-hemorrhagic areas become the same shade of gray, making it difficult to distinguish these areas. If the saturation and brightness remain the same and the colors are distinguished only by the difference in hue, the colors will merge when the saturation is set to 0 and the mode is monochromatic. Hence, we considered that, by including a small amount of saturation, rather than setting the saturation to 0, color distinction would become easier.

In this study, the contrast ratio was increased in the monochromatic mode for ILM staining using BBG. In addition, we showed that, by adding a slight amount of saturation, it became easier to distinguish retinal hemorrhage without significantly changing the contrast ratio. By increasing the contrast ratio by using the monochromatic mode and further adjusting the saturation to distinguish the color tone, ILM peeling could be performed safely.

This study has some important limitations. First, the analysis of hemorrhage detection is based on the small number of cases and should be considered preliminary. Second, clinical outcomes, such as actual surgical performance metrics (e.g., ILM peeling time, ease of procedure, complication rates) or postoperative visual outcomes, were not assessed in this study. Therefore, this work should be considered a proof-of-concept study demonstrating the technical feasibility of contrast enhancement rather than evidence of clinical superiority. Third, contrast measurements were performed by only two evaluators, and interobserver variability was not statistically assessed through formal reliability testing. This may have affected the reproducibility of our contrast measurements. Fourth, although the contrast ratio was increased by using the monochromatic mode, determining whether ILM peeling is actually easier requires evaluation by multiple surgeons. Moreover, the number of cases was small; thus, more cases would need to be included in future studies. Fifth, this study is limited to NGENUITY version 1.4, and the findings may not be generalizable to newer versions. Future studies should validate these findings across different systems and software versions. The NGENUITY ver1.5 is equipped with a mode that emphasizes blue even in the non-monochromatic mode, which will need to be considered in future studies. Further large multicenter cohort studies with clinical outcome measures are required to validate the present findings and establish the clinical benefits of these image enhancement techniques.

## Conclusions

In ILM staining using BBG, using the monochromatic mode with the NGENUITY system improves contrast visibility; however, by adding a small amount of saturation, retinal hemorrhage can be distinguished more easily. In this way, ILM peeling may be performed more safely than by using a completely monochromatic mode.

Future research should evaluate actual surgical performance metrics, such as ILM peeling time, complication rates, and post operative visual outcomes, to establish the clinical benefits of these images enhancement techniques. Additionally, studies with larger sample sizes and multiple surgical centers would help validate these preliminary findings and assess their broader applicability in clinical practice.

## Supporting information

**S1 Data. BBG ILM monochrome mode.**
(XLSX)

## Acknowledgments

The authors are grateful to the staff of the Department of Ophthalmology at Juntendo Urayasu Hospital.

## Author contributions

**Conceptualization:** Saaya Fukase, Yoshihito Sakanishi, Ayumi Usui-Ouchi, Shintaro Nakao, Nobuyuki Ebihara.

**Data curation:** Saaya Fukase, Yoshihito Sakanishi, Keiji Matsumoto.

**Formal analysis:** Yoshihito Sakanishi, Keiji Matsumoto.

**Investigation:** Saaya Fukase, Yoshihito Sakanishi, Keiji Matsumoto.

**Methodology:** Saaya Fukase, Yoshihito Sakanishi, Ayumi Usui-Ouchi, Shintaro Nakao.

**Project administration:** Yoshihito Sakanishi, Nobuyuki Ebihara.

**Resources:** Nobuyuki Ebihara.

**Validation:** Saaya Fukase, Keiji Matsumoto.

**Visualization:** Saaya Fukase.

**Writing – original draft:** Saaya Fukase, Yoshihito Sakanishi.

**Writing – review & editing:** Saaya Fukase, Yoshihito Sakanishi, Ayumi Usui-Ouchi, Shintaro Nakao, Nobuyuki Ebihara.

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
