## [Decision Letter · Decision Letter 0]

22 Sep 2025

Dear Dr. Sakanishi,

Thank you for submitting your manuscript to PLOS ONE. After careful consideration, we feel that it has merit but does not fully meet PLOS ONE’s publication criteria as it currently stands. Therefore, we invite you to submit a revised version of the manuscript that addresses the points raised during the review process.

We look forward to receiving your revised manuscript.

Kind regards,

Koichi Nishitsuka

Academic Editor

PLOS ONE

Journal Requirements:

2. Please provide additional details regarding participant consent. In the ethics statement in the Methods and online submission information, please ensure that you have specified what type you obtained (for instance, written or verbal, and if verbal, how it was documented and witnessed). If your study included minors, state whether you obtained consent from parents or guardians. If the need for consent was waived by the ethics committee, please include this information

For additional information about PLOS ONE ethical requirements for human subjects research, please refer to http://journals.plos.org/plosone/s/submission-guidelines#loc-human-subjects-research .

3. We note that your Data Availability Statement is currently as follows:

“All relevant data are within the manuscript and its Supporting Information files.”

Additional Editor Comments:

In addition to the reviewer’s comments, please address the following points in your revision:

1) The analysis of hemorrhage detection is based on only a few cases and should be considered preliminary. 

   Please acknowledge this explicitly in the Discussion as a limitation.  

2) Clinical outcomes such as surgical performance or postoperative vision were not assessed, which limits the study to proof-of-concept. 

   Please add a statement to this effect in the Discussion.  

3) Contrast measurements were performed by two evaluators, but inter-observer variability was not statistically assessed. 

   Please note this in the limitations section.

Reviewer #1:

Academic Editor:

1) The analysis of hemorrhage detection is based on only a few cases and should be considered preliminary. Please acknowledge this explicitly in the Discussion as a limitation.

2) Clinical outcomes such as surgical performance or postoperative vision were not assessed, which limits the study to proof-of-concept. Please add a statement to this effect in the Discussion.

3) Contrast measurements were performed by two evaluators, but inter-observer variability was not statistically assessed. Please note this in the limitations section.

Reviewers' comments:

Reviewer's Responses to Questions

**Comments to the Author**

1. Is the manuscript technically sound, and do the data support the conclusions?

Reviewer #1: Yes

2. Has the statistical analysis been performed appropriately and rigorously?

Reviewer #1: Yes

3. Have the authors made all data underlying the findings in their manuscript fully available?

Reviewer #1: Yes

4. Is the manuscript presented in an intelligible fashion and written in standard English?

Reviewer #1: Yes

Reviewer #1: This study is noteworthy as it explores an underexamined area of digitally assisted vitreoretinal surgery and provides a proof-of-concept that slight adjustments in saturation can enhance intraoperative safety by facilitating detection of retinal hemorrhage while preserving ILM contrast. I recommend the authors consider the following points:

1. Although not the primary aim of this study, the authors could suggest that future research evaluate actual surgical performance or clinical outcomes (e.g., ILM peeling time, ease of procedure, complication rates, postoperative vision).

2. The analysis of hemorrhage detection appears to rely on a limited number of cases; statistical robustness is therefore limited. Moreover, no quantitative data on the frequency or severity of hemorrhage across groups are provided.

3. The study is restricted to NGENUITY version 1.4, and the findings may not necessarily be generalizable to newer versions or other visualization platforms.

4. Contrast measurements were performed by only two evaluators, and potential inter-observer variability was not fully assessed.

**Do you want your identity to be public for this peer review?** For information about this choice, including consent withdrawal, please see our Privacy Policy

Reviewer #1: **Yes: ** Hamid Riazi-Esfahani

---

## [Author Response · Author response to Decision Letter 1]

30 Oct 2025

EDITOR COMMENTS

Editor Comment 1:

"The analysis of hemorrhage detection is based on only a few cases and should be considered preliminary. Please acknowledge this explicitly in the Discussion as a limitation."

Response: We agree with this important concern regarding the limited sample size for hemorrhage detection analysis. We have explicitly acknowledged this limitation and emphasized the preliminary nature of these findings in our Discussion section.

Changes made: Discussion section, Limitations paragraph - Added text:

"First, the analysis of hemorrhage detection is based on the small number of cases and should be considered preliminary."

Editor Comment 2:

"Clinical outcomes such as surgical performance or postoperative vision were not assessed, which limits the study to proof-of-concept. Please add a statement to this effect in the Discussion."

Response: We acknowledge that our study did not assess clinical outcomes and have explicitly stated that this work should be considered a proof-of-concept study demonstrating technical feasibility rather than clinical superiority.

Changes made: Discussion section, Limitations paragraph - Added text:

"Second, clinical outcomes such as actual surgical performance metrics (e.g., ILM peeling time, ease of procedure, complication rates) or postoperative visual outcomes were not assessed in this study. Therefore, this work should be considered a proof-of-concept study demonstrating the technical feasibility of contrast enhancement rather than evidence of clinical superiority."

Editor Comment 3:

"Contrast measurements were performed by two evaluators, but inter-observer variability was not statistically assessed. Please note this in the limitations section."

Response: We have acknowledged this limitation and noted that the lack of formal inter-observer variability assessment may affect the reproducibility of our contrast measurements.

Changes made: Discussion section, Limitations paragraph - Added text:

"Third, contrast measurements were performed by only two evaluators, and interobserver variability was not statistically assessed through formal reliability testing. This may affect the reproducibility of our contrast measurements."

REVIEWER #1 COMMENTS

Reviewer Comment 1:

"Although not the primary aim of this study, the authors could suggest that future research evaluate actual surgical performance or clinical outcomes (e.g., ILM peeling time, ease of procedure, complication rates, postoperative vision)."

Response: We appreciate this excellent suggestion and have incorporated specific directions for future research that include clinical outcome measures to validate our technical findings.

Changes made: Conclusion section - Added text:

"Further large multicenter cohort studies with clinical outcome measures are required to validate the present findings and establish the clinical benefits of these image enhancement techniques."

Reviewer Comment 2:

"The analysis of hemorrhage detection appears to rely on a limited number of cases; statistical robustness is therefore limited. Moreover, no quantitative data on the frequency or severity of hemorrhage across groups are provided."

Response: We acknowledge this limitation regarding the statistical robustness of our hemorrhage detection analysis. This concern has been addressed through the addition of explicit acknowledgment of the preliminary nature of these findings.

Changes made: Discussion section, Limitations paragraph - (Addressed through the same addition as Editor Comment 1 response, acknowledging the preliminary nature of hemorrhage detection analysis based on small sample size.)

Reviewer Comment 3:

"The study is restricted to NGENUITY version 1.4, and the findings may not necessarily be generalizable to newer versions or other visualization platforms."

Response: We fully agree with this important limitation regarding generalizability. We have acknowledged that our findings are specific to NGENUITY version 1.4 and have noted the need for validation across different systems and software versions.

Changes made: Discussion section, Limitations paragraph - Added text:

"Fifth, this study is limited to NGENUITY version 1.4, and the findings may not be generalizable to newer versions. Future studies should validate these findings across different systems and software versions. The NGENUITY ver1.5 is equipped with a mode that emphasizes blue even in the non-monochromatic mode, which will need to be considered in future studies."

Reviewer Comment 4:

"Contrast measurements were performed by only two evaluators, and potential inter-observer variability was not fully assessed."

Response: This important methodological limitation has been addressed in our revised manuscript through explicit acknowledgment in the limitations section.

Changes made: Discussion section, Limitations paragraph - (Addressed through the same addition as Editor Comment 3 response, noting the lack of statistical assessment of inter-observer variability.)

---

## [Editor Report · Decision Letter 1]

10 Nov 2025

Relationship between saturation and contrast during internal limiting membrane peeling in 3D digital vitreoretinal surgery: A retrospective study

PONE-D-25-33091R1

Dear Dr. Sakanishi,

We’re pleased to inform you that your manuscript has been judged scientifically suitable for publication and will be formally accepted for publication once it meets all outstanding technical requirements.

Kind regards,

Koichi Nishitsuka

Academic Editor

PLOS ONE
---

## [Editor Report · Acceptance letter]

PONE-D-25-33091R1

PLOS ONE

Dear Dr. Sakanishi,

I'm pleased to inform you that your manuscript has been deemed suitable for publication in PLOS ONE. Congratulations! Your manuscript is now being handed over to our production team.

Kind regards,

on behalf of

Dr. Koichi Nishitsuka

Academic Editor

PLOS ONE